# Features of Photosynthesis in *Arabidopsis thaliana* Plants with Knocked Out Gene of Alpha Carbonic Anhydrase 2

**DOI:** 10.3390/plants12091763

**Published:** 2023-04-25

**Authors:** Elena M. Nadeeva, Lyudmila K. Ignatova, Natalia N. Rudenko, Daria V. Vetoshkina, Ilya A. Naydov, Marina A. Kozuleva, Boris N. Ivanov

**Affiliations:** Institute of Basic Biological Problems, Federal Research Center “Pushchino Scientific Center for Biological Research of the Russian Academy of Sciences”, 142290 Pushchino, Moscow Region, Russia; zhurikova-alena@yandex.ru (E.M.N.); lkign@rambler.ru (L.K.I.); nataliacherry413@gmail.com (N.N.R.); vetoshkina_d@mail.ru (D.V.V.); eliotfur@gmail.com (I.A.N.); kozuleva@gmail.com (M.A.K.)

**Keywords:** carbonic anhydrase, *Arabidopsis thaliana*, photosynthesis, thylakoid membrane, protons, non-photochemical quenching

## Abstract

The knockout of the *At2g28210* gene encoding α-carbonic anhydrase 2 (α-CA2) in *Arabidopsis thaliana* (Columbia) led to alterations in photosynthetic processes. The effective quantum yields of both photosystem II (PSII) and photosystem I (PSI) were higher in α-carbonic anhydrase 2 knockout plants (α-CA2-KO), and the reduction state of plastoquinone pool was lower than in wild type (WT). The electron transport rate in the isolated thylakoids measured with methyl viologen was higher in α-CA2-KO plants. The amounts of reaction centers of PSII and PSI were similar in WT and α-CA2-KO plants. The non-photochemical quenching of chlorophyll *a* fluorescence in α-CA2-KO leaves was lower at the beginning of illumination, but became slightly higher than in WT leaves when the steady state was achieved. The degree of state transitions in the leaves was lower in α-CA2-KO than in WT plants. Measurements of the electrochromic carotenoid absorbance shift (ECS) revealed that the light-dependent pH gradient (ΔpH) across the thylakoid membrane was lower in the leaves of α-CA2-KO plants than in WT plants. The starch content in α-CA2-KO leaves was lower than in WT plants. The expression levels of the genes encoding chloroplast CAs in α-CA2-KO changed noticeably, whereas the expression levels of genes of cytoplasmic CAs remained almost the same. It is proposed that α-CA2 may be situated in the chloroplasts.

## 1. Introduction

Carbonic anhydrase (CA) is an enzyme that catalyzes the hydration of carbon dioxide (CO_2_ + H_2_O → HCO_3_^−^ + H^+^) and the dehydration of bicarbonate (HCO_3_^−^ + H^+^ → CO_2_ + H_2_O). The rate constant of the catalyzed reaction is up to 10^6^ times higher than that of the spontaneous reaction [1], with the hydration reaction being accelerated to a greater extent. Several CAs have been found in chloroplasts where photosynthetic reactions occur: α-CA1 [2] and two forms of the soluble CA, β-CA1 [3,4], are located in the stroma, while α-CA4 [5,6] and α-CA5 [7] are bound to the thylakoid membrane. A β-family CA was also discovered in the thylakoid lumen of pea and Arabidopsis [8,9]. However, the specific functions of most of these CAs in photosynthesis have not yet been fully elucidated [10].

Ribulose-1,5-bisphosphate carboxylase-oxygenase (Rubisco) is the key enzyme of the Calvin-Benson cycle that incorporates CO_2_ molecules into organic compounds, such as carbohydrates. This enzyme is located in the chloroplast stroma, where bicarbonate is the main form of inorganic carbon. It was long believed that soluble stromal β-CA1 directly converts bicarbonate to CO_2_ and supplies it to Rubisco. However, this assumption has been challenged by experimental results [10], suggesting that the role of β-CA1 in the chloroplast carbon metabolism is more complex and requires further investigation.

Recent mass spectrometry analysis identified α-CA5 as the only putative carbonic anhydrase located in stromal thylakoid membranes enriched with complexes of PSI [7]. In the experiments with Arabidopsis and pea, α-CA5 was found to be responsible for the increase in the rate of photophosphorylation with the increase in the bicarbonate concentration in the suspension of thylakoids [7,11], a phenomenon discovered many years ago [12].

The proteins of the PSII core complexes as well as the low-molecular-mass proteins of granal thylakoid membranes have been found to possess CA activity [8,13,14,15]. While it has not been determined which CA is related to the CA activity of the PSII core complex [13], α-CA4, which had earlier been found among the proteins of the thylakoid membranes [5], has been identified as one of the low-molecular-mass proteins responsible for the CA activity of granal thylakoid membranes [6,15]. In the leaves of Arabidopsis plants with the α-CA4 gene knocked out (α-CA4-KO), the effective quantum yield of PSII at saturating light intensity and high CO_2_ concentration (800 ppm) was higher than in WT plants, while the non-photochemical quenching of chlorophyll *a* fluorescence (NPQ) was lower [16]. Lower NPQ values were attributed to a lower energy-dependent component of NPQ (qE) [16,17,18]. Despite that, the mutant plants had a higher content of the PsbS protein, which is involved in the development of qE, than WT plants [19]. In the absence of α-CA4, there was no change in the rate of electron transport through PSI in the thylakoids from the mutants compared with WT thylakoids. α-CA4 was found to be located near PSII on the lumenal side of the thylakoid membrane [6,15]. Based on the observed effects of α-CA4 absence, it was hypothesized that this CA supplies protons directly to the PsbS protein [19,20,21], protecting PSII from photoinhibition due to an increase in qE. Additionally, mutant plants exhibited a 4–5 times greater accumulation of starch in their leaves compared to WT plants during the illumination period following the night period [16].

The exact localization of α-CA2 in Arabidopsis is still unclear [3]. However, it has been shown that under low CO_2_ concentration in the air (150 ppm) and at a light intensity of 300 µmol quanta m^−2^ s^−1^, the expression of the gene encoding α-CA2 is significantly upregulated compared to plants grown at ambient CO_2_ levels (360 ppm) [3]. Moreover, we have observed that the expression level of this gene increases with increasing light intensity [20]. Furthermore, a knockout of the gene encoding this CA led to an increase in the rate of CO_2_ assimilation in leaves under high light intensity compared to WT plants [21], indicating a potential role of α-CA2 in photosynthesis.

This paper presents experimental results suggesting that α-CA2 is presumably located in chloroplasts, possibly in thylakoid membranes.

## 2. Materials and Methods

### 2.1. Plant Material and Growth Conditions

Plants of *Arabidopsis thaliana* of the Columbia ecotype and plants with knocked out gene *At2g28210* encoding α-CA2 (homozygous line 9–11, derived from the SALK_120400 line, and homozygous line 8–3, derived from the SALK_080341C line) were used. The plant seeds were kindly provided by Prof. J.V. Moroney (Louisiana State University, USA). The seeds of each of the plant genotypes were sown in three separate pots with soil and placed in a climatic chamber at a constant temperature of 18–20 °C, illumination of 50 μmol quanta m^−2^ s^−1^ (24 h), and CO_2_ concentration of 400 ppm. After seed germination, the plants were grown under illumination conditions of 50 μmol quanta m^−2^ s^−1^ 8 h day/16 h night. The conditions of growth were not changed further on. After 14–21 days, at the time of the formation of four true leaves, plants were transplanted into individual pots with a soil volume of 150 mL. For the experiments, 45–55 days old plants were used. The appearance of the WT and mutant plants is shown in Appendix A. The weights of the aboveground parts of WT plants and mutant plants are shown in Appendix A.

### 2.2. Measurement of Chlorophyll a Fluorescence and P700 Absorption Changes at Room Temperature

Parameters of chlorophyll *a* fluorescence were measured in attached leaves, using a DUAL PAM-fluorometer (Walz, Effeltrich, Germany). Prior to measurement, plants were dark-adapted for 2 h. The measurements were made at an actinic light intensity of 530 μmol quanta m^−2^ s^−1^ from an intrinsic source. Saturating light pulses (0.8 s, 8000 μmol quanta m^−2^ s^−1^) were applied after 7 min of illumination when the steady state level of fluorescence had been achieved. The effective quantum yield of PSII (YII), the parameter 1-qL, characterizing the redox state of the plastoquinone (PQ) pool, and the coefficient of non-photochemical quenching of chlorophyll *a* fluorescence (qN) were calculated as: YII = (F_m_’−F_S_)/F_m_’, 1-qL =1−((F_m_’−F_s_)/(F_m_’−F’_0_)) × F’_0_/F_s_, qN = 1−(F_m_’−F_0′_)/(F_m_−F_0_), where F_m_ is the maximum fluorescence yield in response to a saturating pulse applied to dark-adapted leaves, F_0_ is the minimal fluorescence level in the dark-adapted state, F’_0_ is the minimal fluorescence level in the light-adapted state, F_m_′ is the maximum fluorescence yield in response to a saturating pulse under illumination, and F_s_ is the steady-state fluorescence level. The parameter NPQ, which characterizes non-photochemical quenching of chlorophyll *a* fluorescence according to Stern-Folmer approach and does not depend on F’_0_ under illumination, was calculated as NPQ = (F_m_−F_m_’)/F_m_’ [22].

To evaluate the photoinhibition of PSII, plants were exposed to light with an intensity of 530 µmol quanta m^−2^ s^−1^ for 3 and 6 h, which was the same intensity used to evaluate the photosynthetic characteristics of WT and mutant plants. The ratio (F_m_–F_0_)/F_0_ was used to assess photoinhibition. This ratio was found to be more sensitive to stress factors on plants than the commonly used ratio (F_m_–F_0_)/F_m_ [23]. The ratio (F_m_–F_0_)/F_0_ provides higher dynamic range and more distinctly reflects changes in (F_m_–F_0_) and/or F_0_. A decrease in this ratio indicates deterioration or even damage of the PSII.

The saturation pulse method was used to determine the effective quantum yield of PSI (YI), which was calculated as: Y(I) = (P_m_’–P)/P_m_. Quantum yields of non-photochemical energy dissipation due to donor side limitation Y(ND) and due to acceptor side limitation Y(NA) were calculated as (P_m–_P_m_’)/P_m_ and as P/P_m_, respectively [24]. P_m_, the maximal P700 signal, was determined by the application of the saturation pulse over far-red light in dark-adapted leaves; P was determined when complete reduction of P700 had been induced after the saturation pulse and cessation of far-red illumination; P_m_’ is the maximal P700 signal induced by combined actinic illumination plus the saturation pulse.

To measure the OJIP chlorophyll *a* fluorescence transient in leaves, a Handy Pea fluorometer (Hansatech, King’s Lynn, Great Britain) was used to record the fluorescence during a 1 s flash of red light of 3000 μmol quanta m^−2^ s^−1^ (Appendix A). The measurements were conducted on plants that had been dark-adapted for 2 h. From these measurements, the parameters S_m_ and PI_total_, which reflect the redox state of the chloroplast plastoquinone pool and a total photosynthetic performance index, respectively, were calculated as in [25].

### 2.3. Measurements of Electron Transport Rate in Thylakoids

Thylakoids were isolated from leaves of 5 to 6-week-old Arabidopsis plants of both WT and α-CA2-KO using the method described in [26]. The light-induced rate of photosynthetic electron transfer with methyl viologen (MV), as an electron acceptor receiving electrons from terminal acceptors of PSI, was measured as the rate of oxygen consumption using a custom-built Clark-type pO_2_-electrode in a temperature-controlled glass cell at 21 °C. Illumination of 500 μmol quanta m^−2^ s^−1^ was provided using a light-emitting diode (Epistar, Hsinchu, Taiwan) with a peak wavelength of 660 nm. The reaction medium contained 0.1 M sucrose, 20 mM NaCl, 5 mM MgCl_2_, 50 µM MV, 50 mM HEPES-KOH (pH 7.6), and thylakoids with a chlorophyll concentration of 20 µg Chl mL^−1^.

### 2.4. Evaluation of State Transitions

The transition from state 1 to state 2 was evaluated in separated leaves of the α-CA2-KO and WT plants by recording the low-temperature (77 K) fluorescence spectra of Chl *a*. Chlorophyll fluorescence was excited in a frozen leaf by 435 nm light, and the fluorescence spectra were recorded from 650 to 800 nm with a Hitachi 850 spectrofluorometer (Hitachi, Tokyo, Japan). The PSI/PSII ratio, which represents the ratio of fluorescence peaks of chlorophyll *a* associated with PSI (745 nm) and PSII (685 nm) [27], was measured immediately after 90 min of dark adaptation of leaves, as the control, when the antenna proteins were dephosphorylated and the entire LHCII was connected to PSII. To induce the transition to state 2, leaves were illuminated for 20 min before freezing using a diode with a maximum emission at 640 ± 10 nm, which predominantly excites PSII. The intensity and duration of the illumination were based on previous studies conducted with Arabidopsis plants [28].

The process of state transitions is one of the mechanisms of NPQ, and it can be estimated at room temperature by measuring the relaxation of the NPQ in darkness following illumination. Vetoshkina et al. [28] observed that in attached leaves of Arabidopsis, the relaxation of the NPQ part related to the transition from state 2 to state 1 occurs at room temperature between the 15th and 24th min of darkness after switching off the actinic light that mainly excites PSII.

### 2.5. Electrochromic Carotenoid Absorbance Shift Measurements

The signal of electrochromic carotenoid absorbance shift (ECS) arises when illumination causes electron transfer in PETC, which leads to the electric field building up across the thylakoid membrane, affecting the absorption spectrum of carotenoids located there. ECS was monitored on attached leaves at 515 nm using Dual-PAM-100, equipped with the P515/535 emitter-detector module (Walz, Effeltrich, Germany). Plants were dark-adapted for 90 min, and single-turnover flashes of 2.5 μs duration, alternating with 5 s dark interval, were applied to excite electron transfer in the reaction centers of PSI and PSII. The averaged signal from 50 flashes was accumulated in the Fast Kinetics mode. The maximum amplitude of the averaged signal was calculated and then used as the ECS_st_ value. The determination of pmf and its components, ΔpH and ΔΨ, was performed on dark-adapted leaves after 1 min of illumination and in the same leaves after illumination for 7 min with 530 μmol photons m^−2^ s^−1^, followed by a dark interval for 45–60 s. The value of pmf was determined from the difference between the values of the ECS signal in the light at the moment of light switching off (ECS_t_) and the minimum of the inverted signal in the dark after light switching off. The difference values were normalized to the ECS_st_ determined for a given leaf. The value of ΔpH was found from the difference in the ECS signal value corresponding to the minimal inverted dark signal and the maximum signal after relaxation of the inverted signal. This difference was also normalized to the ECS_st_. The value of ΔΨ was found as the difference between the values of pmf and ΔpH.

### 2.6. Western Blot Analysis

Immunoblotting was performed as described in the protocol of BioRad laboratories. Protein samples from thylakoid membranes isolated from leaves of 5 to 6-week-old Arabidopsis plants of WT or α-CA2-KO were separated by electrophoresis in a 16% polyacrylamide gel under denaturing conditions using the Mini-PROTEAN Cell system (BioRad, Hercules, CA, USA). Precision Plus Protein Kaleidoscope (10–250 kDa) (BioRad, Hercules, CA, USA) was used as protein molecular weight markers. After electrophoresis, the proteins were transferred onto a PVDF membrane (BioRad, Hercules, CA, USA) using a wet blotting system Mini Trans-Blot Cell (BioRad, Hercules, CA, USA). Immunoblotting was carried out using primary polyclonal rabbit antibodies against D1 and PsaC proteins (Agrisera, Vännäs, Sweden) and secondary goat anti-rabbit IgG, AP conjugated antibodies (BioRad, Hercules, CA, USA). The membranes were visualized using an alkaline phosphatase conjugate substrate kit (BioRad, Hercules, CA, USA). The PVDF membranes were scanned on a flatbed scanner in transmission mode for further analysis. Quantification of the optical density of the bands on the blots was performed in the ImageJ software. The results of the Western-blot analysis were obtained from two independent experiments.

### 2.7. Determination of Starch Content

To measure the starch content in the leaves, the leaves were cut in the morning after 3 h of illumination with 50 μmol quanta m^−2^ s^−1^ or 400 µmol quanta m^−2^ s^−1^. Determination of the starch content was carried out by measuring the optical density at 620 nm in aqueous leaf extracts after incubation with KI [29].

### 2.8. Quantitative Reverse Transcription PCR

Total RNA was extracted from frozen Arabidopsis leaves, using the Aurum total RNA Mini kit (BioRad), and treated with DNase to eliminate any genomic DNA contamination. Complementary DNA synthesis was performed using the reverse transcription kit OT-1 (Sintol, Moscow, Russia) with oligo (dT) as a primer. Quantitative reverse transcription polymerase chain reaction (qRT-PCR) was performed with qPCRmix-HS SYBR (Evrogen, Moscow, Russia), using primer sequences for the genes encoding CAs based on [21], with primers for βca1.1+1.2 corresponding to βca1a primers. The resulting qRT-PCR data were normalized to the housekeeping gene for actin 7. The PCR reactions were carried out in a LightCycler 96 Instrument (Roche Diagnostics, Mannheim, Germany).

### 2.9. Statistical Analysis

Statistical analyses were performed using OriginPro, 2021 (OriginLab Corporation, Northampton, MA, USA). The results presented in figures and tables are expressed as mean values ± SE. ANOVA analysis was performed for all figures and tables, followed by a Holm-Bonferroni test for pairwise comparison of means. Statistical significance was denoted as: * *p* ≤ 0.05, ** *p* ≤ 0.10, *** *p* ≤ 0.15.

## 3. Results

### 3.1. Effect of α-CA2 Absence on the Photosynthetic Electron Transport in Intact Leaves

We measured the effective quantum yields YII and YI in the leaves of WT plants and α-CA2-KO mutants, specifically lines 9–11 and 8–3. These measurements were taken in high light using the classical scheme with application of saturating pulses over the actinic light. Both YII and YI in α-CA2-KO mutants were higher compared to WT (Figure 1A,C). The Y(ND) parameter, which characterizes the donor-side limitation of PSI, was lower in α-CA2-KO mutants than in WT plants. On the other hand, the Y(NA) parameter, which characterizes the limitation of the PSI acceptor side, did not differ significantly between α-CA2-KO mutants and WT plants. These results indicate a higher rate of electron transport in α-CA2-KO mutants compared to WT plants [30,31]. Furthermore, measurements of the 1-qL parameter, which approximately characterizes the redox state of the plastoquinone (PQ) pool [32], under the same conditions revealed that this pool was more oxidized in mutants than in WT plants (Figure 1B).

The electron transfer through PETC and the redox state of PQ pool in the leaves of WT and α-CA2-KO plants were also compared using measurements of the OJIP kinetics of chlorophyll *a* fluorescence. The PI_total_ parameter, the total plant performance index, which characterizes the performance of the total electron flux, was higher in mutant plants than in WT plants (Figure 2A). The S_m_ parameter, which characterizes the relative reduction level of the PQ pool [33], was higher in both mutant lines than in WT plants, indicating a lower level of PQ pool reduction in α-CA2-KO than in WT (Figure 2B). This was consistent with the conclusion drawn from the measurement of the 1-qL coefficient (Figure 1B).

### 3.2. Effect of α-CA2 Absence on the Electron Transport in Isolated Thylakoids

The electron transfer rate in isolated thylakoids was measured as the rate of oxygen uptake in the presence of MV, an auto-oxidizable acceptor that is reduced at the acceptor side of PSI, i.e., the rate of electron transfer along the entire PETC from water (Table 1). The rate of electron transport was limited by pH lumen in all cases, as evidenced by stimulation of the electron transfer rate in the presence of gramicidin D, i.e., under uncoupling conditions, in thylakoid preparations from WT and both mutant lines (data not shown). The measurements of the photosynthetic electron transport in thylakoids isolated from the leaves of WT and α-CA2-KO plants (Table 1) were consistent with the results obtained with intact leaves (Figure 1A,C), indicating that the electron transfer rate in α-CA2-KO plants was higher than in WT.

### 3.3. Effect of α-CA2 Absence on the Content of Photosynthetic Reaction Centers Proteins

To determine whether the observed differences between α-CA2-KO and WT in the efficiency of electron transfer through PSII and PSI were caused by the different amounts of PSI or PSII reaction centers, we measured the content of the proteins of PSII, D1 (PsbA), and of PSI, PsaC. The contents of D1 and PsaC proteins were the same in α-CA2-KO and WT plants (Figure 3A,B). Thus, the observed differences between mutants and WT (Figure 1 and Figure 2, and Table 1) have not been determined by the changes in the amount of the reaction centers of either PSII or PSI.

### 3.4. Effect of α-CA2 Absence on Non-Photochemical Quenching of Chlorophyll Fluorescence

The non-photochemical quenching of chlorophyll *a* fluorescence was estimated by the coefficient qN. It was measured in attached leaves during the induction of photosynthesis (after 1 min of illumination) and during steady-state photosynthesis (after 7 min of illumination). At the first minute of high light illumination, the qN coefficient in α-CA2-KO was lower than in WT plants (Figure 4A). In steady-state photosynthesis, the qN coefficient in α-CA2-KO became slightly higher compared with WT (Figure 4B).

### 3.5. Effect of α-CA2 Absence on the Susceptibility to Photoinhibition

The continuous illumination of plants with 530 µmol quanta m^−2^ s^−1^ for 3 h did not lead to a significant difference in the (F_m_–F_0_)/F_0_ ratio between WT plants and α-CA2-KO (Figure 5). However, after 6 h of illumination, the (F_m_–F_0_)/F_0_ ratio in the leaves of α-CA2-KO decreased less compared to WT plants (Figure 5), suggesting that the mutants were more resistant to prolonged exposure to high light.

### 3.6. Effect of α-CA2 Absence on State Transitions

Changes in the redox state of the PQ pool can affect the activation of the STN7 kinase, an enzyme located in the thylakoid membrane and involved in state transitions. State transitions are the reversible migrations between photosystems of mobile loosely bound trimers (L-trimers) of LHCII phosphorylated by active STN7 kinase. This leads to the restoration of the excitation balance between the two photosystems if one of them is predominantly excited.

The transition from state 1 to state 2 [34] was assessed by measuring the low-temperature fluorescence of chlorophyll *a*, as described in Materials and Methods. During the transition to state 2 L-trimers detached from the PSII, leading to a decrease of the fluorescence level of PSII, and the fluorescence level of PSI increased since L-trimers became bonded to the PSI. This leads to an increase of the PSI/PSII fluorescence ratio. The transition from state 1 to state 2 was evaluated by comparing the PSI/PSII ratio before and after illumination. The increase in the PSI/PSII ratio after illumination was 22% in WT plants, 13% in α-CA2-KO plants of line 9–11, and 8% in plants of line 8–3 (Table 2), indicating that the mutants had a lower level of state 1 to state 2 transition.

To estimate the processes of state transitions at room temperature, we evaluated the relaxation of the part of the NPQ (NPQ15′-NPQ24′, see Materials and Methods) associated with the return of the antenna from PSI to PSII, which is the state 2 to state 1 transition. In both lines of the α-CA2-KO mutants, the relaxation of that part of the NPQ upon turning off the actinic light was noticeably lower compared to WT plants (Figure 6). It should be noted that both the transitions from state 1 to state 2 and from state 2 to state 1 were approximately 40% less in mutants than in WT.

### 3.7. Effect of α-CA2 Absence on the Characteristics of Electrochromic Shift in Intact Leaves

Measurements of ECS were performed after 1 min of illumination, i.e., during induction of photosynthesis; and 7 min of illumination, i.e., during stationary photosynthesis. There were no significant differences in the pmf values between WT and α-CA2-KO plants of both lines, although there was a slight tendency for pmf to be lower in the mutants after 1 min of illumination (Figure 7A). After 1 min of illumination, the values of ΔΨ were slightly higher in α-CA2-KO plants of both lines compared to WT plants (Figure 7B), and after 7 min of illumination this difference increased. At the same time, after 1 min of illumination the values of ΔpH in α-CA2-KO plants of both lines were considerably lower than in WT plants (Figure 7C), and this difference decreased after 7 min of illumination, although the absolute ∆pH values in both mutants remained lower. Such reciprocal changes in ΔΨ and ΔpH can explain the pmf values at 1 and at 7 min of illumination, where pmf at 1 min is lower due to a lower ΔpH, and stationary pmf values are aligned due to the lower ΔpH values in the mutants being compensated by an increase in ΔΨ.

### 3.8. Effect of α-CA2 Absence on the Amount of Starch in Leaves

The starch content in the leaves of WT and α-CA2-KO plants was measured 3 h after the start of illumination following the night period at a light intensity used during growth. The results showed that at 50 µmol quanta m^−2^ s^−1^, the starch content was lower by 30–50% in the leaves of α-CA2-KO mutants compared to the leaves of WT plants (Table 3). This effect was also observed at a higher light intensity, 400 µmol quanta m^−2^ s^−1^, although the difference was less pronounced with a reduction of starch content by 18–30% in the leaves of mutants in comparison with WT (Appendix A).

### 3.9. Effect of α-CA2 Absence on the Expression Levels of the Cytoplasmic and Chloroplast Carbonic Anhydrases

Knockout of the gene encoding α-CA2 has a significant impact on the expression levels of the genes encoding chloroplast CAs, α-CA1, β-CA1, α-CA4, and β-CA5; but, has no effect on the expression levels of cytoplasmic CAs, β-CA2 and β-CA3 (Figure 8).

Furthermore, the knockout of α-CA2 gene had opposing effects on the expression levels of the genes encoding two stromal CAs, αCA1, and βCA1.1+1.2, with *αca1* gene expression being higher and *βca1.1+1.2* expression being lower in α-CA2-KO than in the WT (Figure 8). The expression level of the gene encoding thylakoid α-CA4 was about 3.5 times higher, while the expression of the gene encoding another chloroplast CA, β-CA5 [3], was 60–80% lower in α-CA2-KO plants than in the WT plants.

## 4. Discussion

The α-CA2-KO plants did not show any significant phenotypic differences compared to WT plants (Appendix A). However, the total rosette weights (Appendix A) and the starch content per fresh weight (Table 3) were lower in the mutants than in WT plants. The difference in starch content in the leaves of α-CA2-KO and WT plants measured 3 h after the start of illumination following the night period was more pronounced, ranging from 30 to 50%, under low light illumination; i.e., under low input of light energy, compared to 18–30% in the plants acclimated to high light. The absence of α-CA2 also affected the expression levels of genes encoding chloroplast CAs, which could alter the synthesis of these CAs (Figure 8). Since primary starch biosynthesis occurs in chloroplasts, changes in the amount and operation of chloroplast CAs involved in the reactions of CO_2_ and bicarbonate interconversion could potentially affect starch biosynthesis.

We found a significant decrease in the value of ΔpH across the thylakoid membrane in the leaves of α-CA2-KO plants compared to WT plants after 1 min and to a lesser extent after 7 min of illumination, as measured by ECS (Figure 7). In dark-adapted plants, during the initial phase of illumination, only a subset of ATP synthase complexes and Calvin cycle enzymes in the stroma are activated [35,36], with proton conduction of the thylakoid membrane being the main determinant of ΔpH. However, under steady-state conditions, ΔpH is mainly dependent on ATP-synthase activity, which controls proton outflow from the lumen. Other factors, such as the operation of the Calvin cycle, which consumes ATP, and the accumulation of metabolites in the chloroplast stroma, can also affect ΔpH value. These processes may have masked the potential effect of the absence of α-CA2 on proton conduction of the thylakoid membrane.

The preceding discussion suggests that the lower ΔpH observed in leaves of α-CA2-KO compared to WT plants is indicative of a lower proton concentration, i.e., the higher pH, in the thylakoid lumen of the mutants. However, it is possible that the measured value of ΔpH in leaves could be influenced by the pH of the chloroplast stroma. To address this, we measured the electron transport rates in thylakoids isolated from WT and mutant plants (Table 1) at the same pH values in the outer salt buffer. Under these conditions, the higher pH of only the lumen in α-CA2-KO thylakoids could explain the higher electron transport rates compared to WT. The oxidation of PQH_2_ molecules by the cytochrome *b_6_f* complex, which is the main limiting stage of the electron transfer along PETC and releases protons into the thylakoid lumen, becomes faster with a decrease in proton concentration.

The lower magnitude of ΔpH in the leaves of α-CA2-KO plants can account for the higher quantum yield of the PSII and the higher value of PI_total_ (Figure 1A and Figure 2A), the lower level of PQ pool reduction under illumination (Figure 1B and Figure 2B), and the lower value of Y(ND) compared to WT plants (Figure 1D).

The observed changes in the expression levels of genes encoding only chloroplast CAs (Figure 8) upon knockout of the gene encoding α-CA2 suggests that it may be located in this organelle. This idea is supported by previous literature, indicating that CAs located in the same cellular compartment work in close collaboration [10]. Moreover, the effect of the absence of α-CA2 on both the rate of electron transport in isolated thylakoids (Table 1) and the electrochromic shift parameters, reflecting a process in the thylakoid membrane (Figure 7), suggest that this CA may be located in the thylakoids.

In Figure 4, it is shown that the coefficient of non-photochemical quenching of chlorophyll fluorescence, qN, in α-CA2-KO plants was lower than in WT after 1 min of illumination, while after 7 min of illumination, it becomes higher than in WT plants. The value of qN after 1 min of illumination is mainly determined by the development of energy-dependent NPQ, which is triggered by pH decrease in the lumen [37]. The lower value of qN in α-CA2-KO plants after 1 min of illumination is consistent with the lower ΔpH in these plants at this time. However, the higher value of qN in the leaves of α-CA2-KO compared to WT after 7 min of illumination does not match with a lower ΔpH at this time. We have previously found that the energy-dependent NPQ was strongly dependent on the activity of α-CA4 [17,18,19]. α-CA4, being located in thylakoid membrane [5] close to PSII [6], catalyzes the CO_2_ hydration reaction, presumably supplying protons to the PsbS protein [17,19]. Conformational changes in the PsbS protein result in conformational changes in the proteins of the LHCII, increasing energy dissipation into heat [38]. The expression level of the gene encoding the thylakoid α-CA4 in α-CA2-KO was 3.5 times higher than in WT (Figure 8). Such an increase in expression could lead to a higher amount of α-CA4 in these mutants and an increase in the operation of this CA, leading to an increased magnitude of energy-dependent NPQ and total NPQ in the mutants.

The stimulation of non-photochemical quenching of chlorophyll fluorescence is considered one of the primary ways to protect the pigment apparatus and reaction centers of photosystems from photoinhibition. Thus, the increased resistance of PSII to photoinhibition in α-CA2-KO plants compared to WT plants during long illumination (Figure 5) may be attributed to the higher qN value in the mutant plants.

The state transitions in WT and α-CA2-KO were evaluated by measuring the light-induced changes in the ratio of low-temperature fluorescence peaks at 745 and 685 nm, representing PSI and PSII, respectively (Table 2); and the dark relaxation at room temperature of the part of the NPQ, which is dependent on state transitions (Figure 6). The results indicated that the processes of state transitions is less efficient in α-CA2-KO compared to WT plants. It is known that the docking of PQH_2_ with the Qo site of the cytochrome *b_6_f* complex leads to the activation of STN7 kinase [39,40], which is closely related to this complex. The lower extent of state transitions in the mutants may be due to a more oxidized state of the PQ pool in α-CA2-KO than in WT (Figure 1B and Figure 2B).

Previously it was suggested that α-CA2 is located in the thylakoid membrane based on the observation that the differences in photosynthetic characteristics between WT and α-CA2-KO plants were opposite to those between WT and α-CA4-KO plants [21]. The localization of α-CA4 in the thylakoid membrane was reported by Friso et al. [5] and later confirmed in our experiments [6]. The difference in ECS characteristics between α-CA2-KO and WT plants support the above proposition. Furthermore, the lower ΔpH across the thylakoid membrane in α-CA2-KO plants compared to WT plants suggests as one of possible assumptions that the thylakoid membranes of the mutant plants may have greater proton conductivity than those of WT plants (see above).

There is evidence that CAs function as part of protein macromolecular complexes known as metabolons [41,42]. Studies on Arabidopsis plasma membranes have revealed an interaction between β-CA4 and aquaporins [43], and the research on the chloroplast envelope in higher plants has also shown association between CAs and aquaporins [44,45]. In animal tissues, CAs have been found to participate in delivering protons and/or bicarbonate ions to ion-transport systems in cell membranes [41].

We hypothesize that α-CA2 is structurally and functionally associated with one of the ion channels regulating proton leakage through the thylakoid membrane. How can the reaction catalyzed by CA be used for proton outflow from the lumen? Bicarbonate, the substrate of CAs, is the predominant form of inorganic carbon in the alkaline stroma in the light. This hypothetic association enables the protons coming from the lumen through the ion channel to be used by α-CA2, which catalyzes the dehydration of bicarbonate. The absence of α-CA2 might disturb the normal functioning of the ion channel, leading to uncontrolled proton flow. Our findings on α-CA2-KO plants resemble the properties of Arabidopsis plants overexpressing antiporter KEA3 with a point mutation in the transmembrane domain, DPGRox [46]. DPGRox plants displayed lower ∆pH compared to WT. DPGRox plants showed either no difference or higher quantum yields YII compared to WT [46], which is consistent with the data obtained in α-CA2-KO (Figure 1). The value of non-photochemical quenching was lower in DPGRox plants than in WT at the beginning of illumination, but became close to WT after 5 min of illumination. These observations were explained by the increased thylakoid membrane conductivity for protons in DPGRox [46].

Currently, the study of the exact location of α-CA2 in the chloroplasts, as well as the examination of possible connection between α-CA2 and an ion channel in the thylakoid membrane is underway.

## 5. Conclusions

The knockout of *At2g28210* gene encoding α-carbonic anhydrase 2 (α-CA2) in Arabidopsis plants resulted in significant changes in various plant characteristics. The starch content in the leaves noticeably decreased. In the mutants, if compared with WT, the processes closely related to photosynthesis were altered, namely, the degree of state transitions, the ΔpH values across the thylakoid membrane, and the plastoquinone pool reduction decreased, while quantum yields of photosystems increased. Only the expression levels of genes encoding chloroplast carbonic anhydrases, α-CA1, β-CA1, α-CA4, and β-CA5 changed, while the expression levels of genes encoding cytoplasmic carbonic anhydrases remained unchanged. These findings suggest that α-CA2 may be located in chloroplasts, presumably in the thylakoid membrane.

## Figures and Tables

**Figure 1 plants-12-01763-f001:**
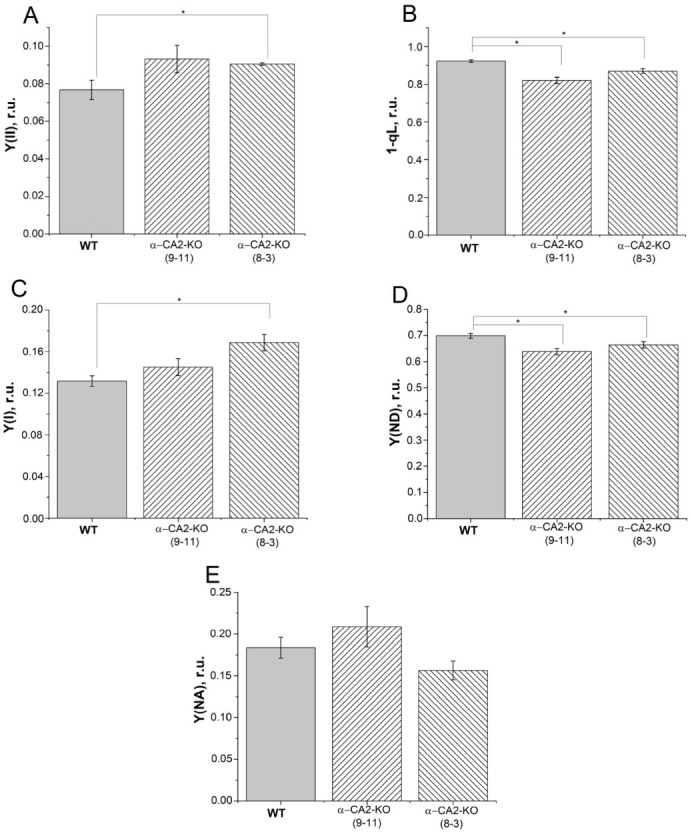
The effect of α-CA2 absence on the parameters of the photosynthetic electron transport in intact leaves. (**A**), the effective quantum yield of PSII, Y(II); (**B**), the coefficient 1-qL; (**C**), the effective quantum yield of PSI, Y(I); (**D**), donor-side limitation of PSI, Y(ND); (**E**), acceptor side limitation of PSI, Y(NA). (The calculations are presented in Methods.) The measurements were performed at light intensity of 530 µmol quanta m^−2^ s^−1^ and CO_2_ concentration of 400 ppm. Gray columns—WT, columns with right shading—α-CA2-KO (9–11 line), columns with left shading—α-CA2-KO (8–3 line). Significant differences are indicated by *, *p* ≤ 0.05.

**Figure 2 plants-12-01763-f002:**
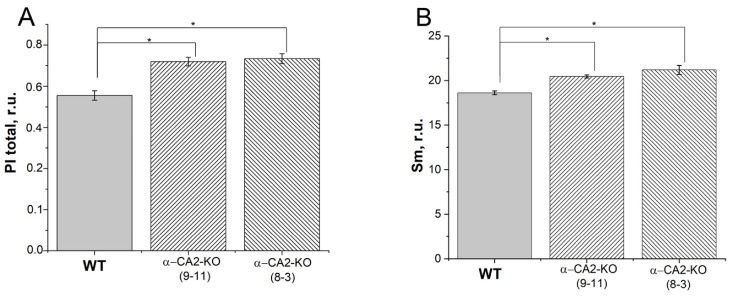
The effect of α-CA2 absence on parameters of OJIP kinetics of chlorophyll *a* fluorescence. (**A**), the total plant performance index, PI_total_; (**B**), the relative reduction level of the PQ pool, Sm. The plants were dark-adapted for 2 h before measurements. Gray columns—WT, columns with right shading—α-CA2-KO (9–11 line), columns with left shading—α-CA2-KO (8–3 line). Significant differences are indicated by *, *p* ≤ 0.05.

**Figure 3 plants-12-01763-f003:**
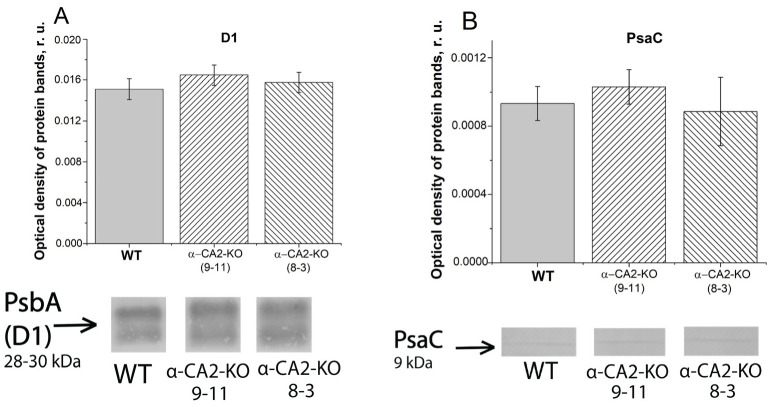
Optical densities of the protein bands obtained as the result of Western blot analysis of thylakoid proteins isolated from leaves of WT and α-CA2-KO plants grown at light intensity of 50 µmol quanta m^−2^ s^−1^, with antibodies against D1 protein of PSII (**A**), and PsaC protein of PSI (**B**). The PVDF membranes with protein bands are shown. Each well was loaded with preparations of the proteins of thylakoid membranes containing 1 μg Chl. Gray columns—WT, columns with right shading—α-CA2-KO (9–11 line), columns with left shading—α-CA2-KO (8–3 line). Complete membranes with protein bands are shown in the Appendix A.

**Figure 4 plants-12-01763-f004:**
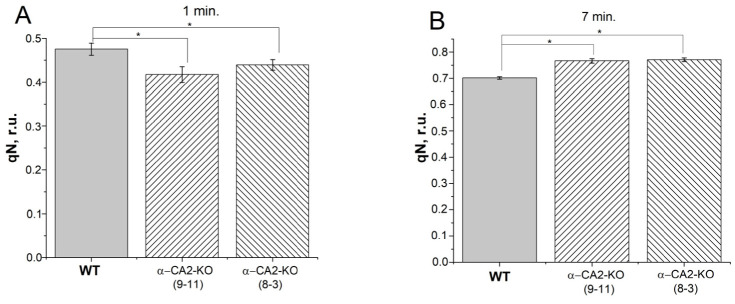
The effect of α-CA2 absence on the coefficient of the non-photochemical quenching of chlorophyll *a* fluorescence, qN, in intact leaves after 1 min of illumination (**A**) and after 7 min of illumination (**B**). (The calculations are presented in Methods.) The measurements were performed at light intensity of 530 µmol quanta m^−2^ s^−1^ and CO_2_ concentration of 400 ppm. Gray columns—WT, columns with right shading—α-CA2-KO (9–11 line), columns with left shading—α-CA2-KO (8–3 line). Significant differences are indicated by *, *p* ≤ 0.05.

**Figure 5 plants-12-01763-f005:**
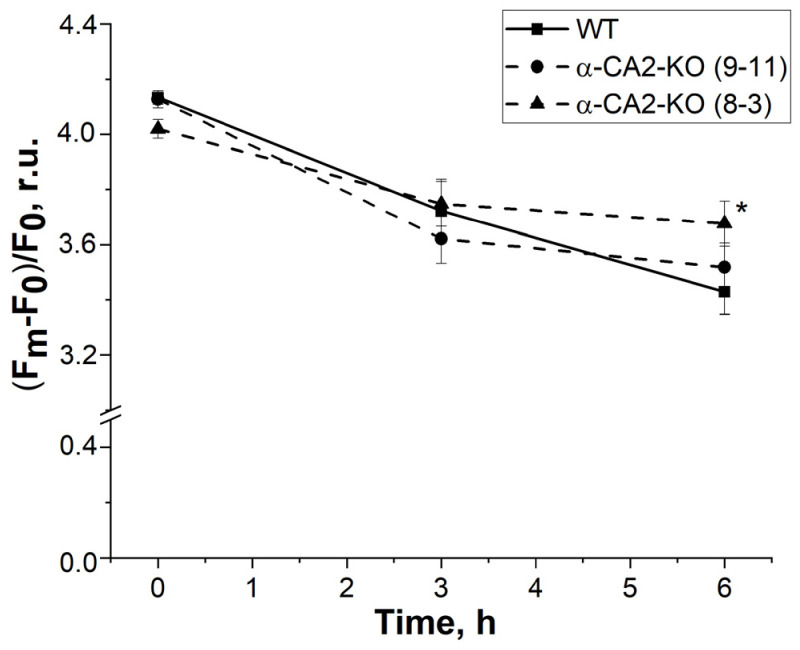
The efficiency of PSII functioning in the leaves of α-CA2-KO (9–11 and 8–3 lines) and WT plants, measured as (F_m_ − F_0_)/F_0_ before and after 3 and 6 h of illumination with light of 530 µmol quanta m^−2^ s^−1^. Significant differences are indicated by *, *p* ≤ 0.05.

**Figure 6 plants-12-01763-f006:**
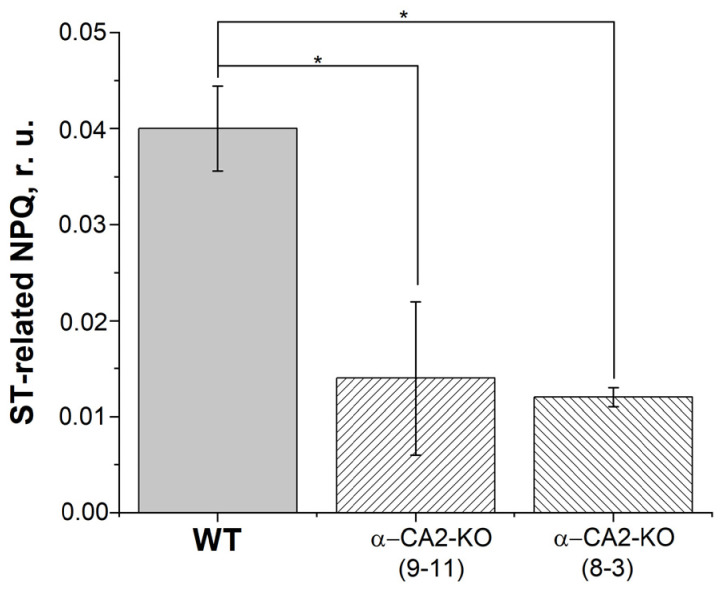
The relaxation of the part of the NPQ associated with state transitions (NPQ15′-NPQ24′, see Materials and Methods) and measured after switching off the light of 60 μmol quanta m^−2^ s^−1^ in the leaves of WT and in plants with knockout of the gene encoding α-CA2, lines 9–11 and 8–3. Plants were grown at 50 μmol quanta m^−2^ s^−1^. Gray columns—WT, columns with right shading—α-CA2-KO (9–11 line), columns with left shading—α-CA2-KO (8–3 line). Significant differences are indicated by *, *p* ≤ 0.05.

**Figure 7 plants-12-01763-f007:**
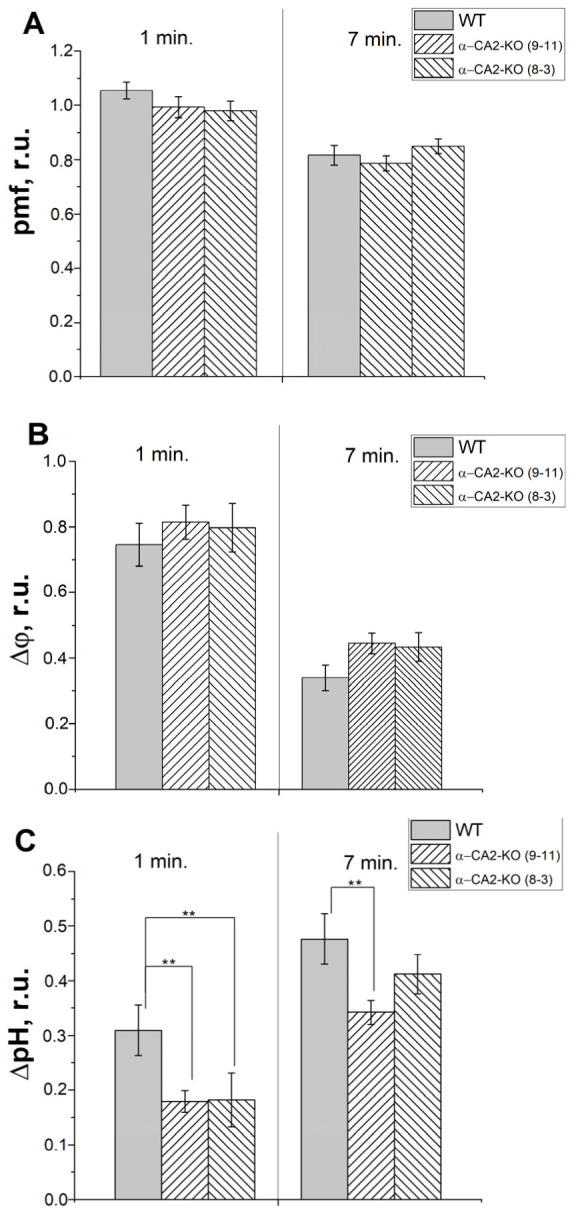
The effect of α-CA2 absence on the proton motive force, pmf (**A**), the difference in electrical potentials, ΔΨ, (**B**), and the difference in pH, ∆pH, (**C**) across the thylakoid membrane after 1 and 7 min of illumination (530 µmol quanta m^−2^ s^−1^). Gray columns—WT, columns with right shading—α-CA2-KO (9–11 line), columns with left shading—α-CA2-KO (8–3 line). Significant differences are indicated by **, *p* ≤ 0.1.

**Figure 8 plants-12-01763-f008:**
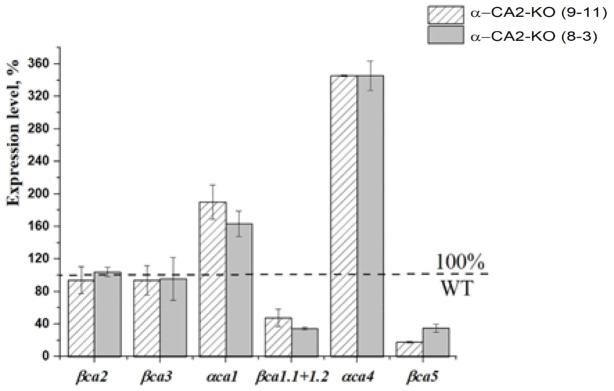
Expression levels of genes encoding cytoplasmic CAs, β-CA2 and β-CA3; and chloroplasts CAs, α-CA1 β-CA1, α-CA4 and β-CA5 in the α-CA2-KO mutant. The expression level values of the corresponding genes in WT plants (dashed horizontal line) were taken as 100%. The results were obtained from two independent experiments.

**Table 1 plants-12-01763-t001:** The rate of electron transport in thylakoids isolated from WT plants and from α-CA2-KO, lines 9–11 and 8–3. The rate of electron transport was measured as the rate of oxygen uptake in the presence of 50 µM MV as an acceptor. Significant differences are indicated by *, *p* ≤ 0.05, ** *p* ≤ 0.10.

Plants	The Electron Transfer Rate, µmol O_2_/mg Chl × h
WT	13.7 ± 1.4
α-CA2-KO (9–11)	16.8 ± 0.9 **
α-CA2-KO (8–3)	18.0 ± 0.8 *

**Table 2 plants-12-01763-t002:** The ratio of the low-temperature chlorophyll *a* fluorescence peaks (PSI/PSII) measured at 77°K from leaves of WT and α-CA2-KO plants (lines 9–11 and 8–3), dark-adapted, or after illumination with red light (λ = 640 nm, 60 µmol quanta m^−2^ s^−1^) for 20 min.

Plants	PSI/PSII Ratio at the 77 K Emission Spectra After Dark Adaptation	PSI/PSII Ratio at the 77 K Emission Spectra After 20 min Illumination with Light (λ = 640 nm)
WT	3.6 ± 0.2 (100%)	4.4 ± 0.1 (122%)
α-CA2-KO (9–11)	3.0 ± 0.2 (100%)	3.4 ± 0.1 (113%)
α-CA2-KO (8–3)	3.6 ± 0.1(100%)	3.9 ± 0.1 (108%)

**Table 3 plants-12-01763-t003:** Starch content in the leaves of WT plants and α-CA2-KO plants (lines 9–11 and 8–3) after 3 h of illumination at a light intensity of 50 µmol quanta m^−2^ s^−1^ following 16 h of dark period.

Plants	Starch Content (mg g^−1^ Fresh Weight)
WT	2.05 ± 0.13
α-CA2-KO (9–11)	1.48 ± 0.13
α-CA2-KO (8–3)	0.92 ± 0.10

## Data Availability

The original research results presented in this paper are not submitted to other journals, and were not previously published.

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
