# Peer review of "Features of Photosynthesis in Arabidopsis thaliana Plants with Knocked Out Gene of Alpha Carbonic Anhydrase 2"

_plants, 2023, doi:10.3390/plants12091763_

Round 1
Author Response
- I wish that the authors have generated an antibody against the alpha CA2 and were able to show the immunolocalization results. This would have strengthened the article.
Thank you for this important comment. Of course, we have such intention, and we indicated this at the end of our manuscript. That is the aim of our future research. For the moment we work on heterologous production of alpha CA2 in the bacterial expression system. In case of successful synthesis of the active enzyme, we plan to obtain antibodies against this protein followed by immunoblotting with the proteins of the fractions isolated from Arabidopsis leaf cells.
Besides, we direct our efforts to prepare the mutant line of Arabidopsis thaliana with knocked-out gene encoding αCA2, using CRISPR/Cas9 genome editing system, as we have already done for alpha-CA4 encoding gene (Plants 2022, 11, 3303. https://doi.org/10.3390/plants11233303). In that study we found that the characteristics of the mutant prepared using CRISPR/Cas9 were similar to those obtained previously with the insertional mutants. What is important, the characteristics of the insertional mutants were not the result of the negative effects of T-DNA insertion transgenesis. We hope for the same result with the insertional mutants with alpha-CA2 knockout that were used in the reviewed study.
- The authors are strongly advised to get the manuscript edited by a professional editor to rectify grammatical mistakes and typos. It was very hard for me to read the manuscript as the grammar problem often changed the meaning conveyed in the sentences.
We worked thoroughly to increase the English language quality. We resorted the help of a professional editor.
- Line73: Maybe a reference is missing?? If not, please delete the extra space.
Thanks, we've added missing reference.
- I would recommend the authors to provide an explanation for the decreased expression of the beta CA1 and CA5 in the chloroplast in the two knockout mutants in the discussion section. . I would recommend the authors to provide an explanation for the decreased expression of the beta CA1 and CA5 in the chloroplast in the two knockout mutants in the discussion section.
We did not discuss the reasons of the decreased expression of the genes encoding beta CA1 and CA5 in the plants with alpha-CA2 knockout since the functions of these CAs are unknown until now. As to function of beta-CA1, the opinions are contradictory. The obvious proposition about the participation of this CA in CO2 delivery to Rubisco in the chloroplast stroma was thought to be refuted in the first studies dedicated to this issue (Badger, Price 1994). This was supported in some other studies. However, there are the facts, which could be interpreted in favor of this function of beta-CA1. Please note, that we indicated the levels of genes encoding the forms βCA1.1+1.2, which is situated in chloroplasts indeed. The detail analysis of the literature and our data concerning the functions of beta-CA1 was presented in the review by Rudenko and Ivanov (2021). The discussion, why the expression level of the gene encoding beta-CA1 decreases in alpha-CA2-KO would be very long ‘caprioles of fancy’.
As to beta-CA5, the situation even more complicated. It is really situated in chloroplasts that was shown in (Fabre et al., 2007; Hines et al., 2021; Weerasooriya et al., 2022). Apparently, this enzyme is a very important for plants, since the knockout of the gene encoding beta-CA5 resulted in a significant suppression of Arabidopsis plants growth (Medina-Puche et al., 2017; Kasili et al., 2023). However, the physiological role of beta-CA5 appears to be unclear. The participation of this enzyme in photosynthesis has not been proven. There is evidence of its involvement in fatty acids synthesis in chloroplasts (Hines et al., 2021), as well as in stress signaling induced by salicylic acid (Medina-Puche et al., 2017). Our studies (Rudenko et al., 2007; Fedorchuk et al., 2014) revealed the presence of a CA in thylakoid lumen, and this CA was shown to have the characteristics of CAs of beta family. We then proposed that this luminal CA is beta CA5. We have unpublished data from mass spectral analysis that this is valid. If our proposition is correct, then it would be easy to make a connection of the function of alpha-CA2 proposed in our manuscript, namely the regulation of proton efflux from lumen with operation of beta-CA5 in lumen. However, it again will be only a speculation.
It seems that it is too early to discuss the reasons of the influence of alpha-CA2 absence on the expression levels of genes encoding beta-CA1 and beta-CA5.
Reviewer 2 Report
Comments to the Author
Title: Features of photosynthesis in Arabidopsis thaliana plants with knocked out gene of alpha carbonic anhydrase 2
In this study, the authors evaluated the effects of α-CA2 absence on photosynthetic electron transport rate, the content of photosynthetic reaction centre proteins, non-photochemical quenching of chlorophyll fluorescence, susceptibility to photoinhibition, state transitions, the characteristics of electrochromic shift, starch content and the expression levels of the cytoplasmic and chloroplast carbonic anhydrases. Though it is an meaningful and substantial study, there are some minor issues that need to be addressed.
Materials and methods:
Line 95-97: You should specify the growth conditions, you only describe the temperature, illumination and CO2 concentration, what is the growth culture of Arabidopsis thaliana before and after transplanting?
Line 123: “ As it was sown in...”? Is it “As it was shown in...”?
Line 90-221: you measured many indexes about photosynthesis, why not measure the Pn, Tr, Gs and Ci? Maybe it is perfect to add these parameters.
Line 223: Please specify the Origin Pro software information(version and country).
Discussion
Line 420-540: This section seems a little redundant, please condense this section.
Author Response
- Line 95-97: You should specify the growth conditions, you only describe the temperature, illumination and CO2 concentration, what is the growth culture of Arabidopsis thaliana before and after transplanting?
We added in the text the description of the growth culture:
The seeds of each of the plant genotypes were sown in three separate pots with soil and placed in a climatic chamber at a constant temperature of 18–20 °C, illumination of 50 μmol quanta m-2 s-1 (24 h), and CO2 concentration of 400 ppm. After seed germination, the plants were grown under illumination conditions of 50 μmol quanta m-2 s-1 8 h day/16 h night. The conditions of growth were not changed further on. After 14–21 days, at the time of the formation of four true leaves, plants were transplanted into individual pots with a soil volume of 150 ml.
- Line 123: “ As it was sown in...”? Is it “As it was shown in...”?
Yes, certainly.
- Line 90-221: you measured many indexes about photosynthesis, why not measure the Pn, Tr, Gsand Ci? Maybe it is perfect to add these parameters.
Thank you for the important comment. We measured the rate of CO2 assimilation in leaves several years ago using the Portable gas exchange and fluorescence system GFS-3000, Walz. The measurements were done only at high light intensity (Zhurikova et al., 2016), and these data we referred in the manuscript. At the present time, we work on creating of new alpha-CA2 knocked-out line, using CRISPR/Cas9 genomic editing (see also reply to Reviewer 1). We also possess LI-6800 Portable Photosynthesis System (LiCor) which will allow us to measure all the specified parameters (Pn, Tr, Gs and Ci). Taking into account your advice, we are intend to obtain in the future research the indicated parameters using CRISPR/Cas9 mutant.
- Line 223: Please specify the Origin Pro software information (version and country).
Statistical analyses were conducted using OriginPro, 2021 (OriginLab Corporation, Northampton, MA, USA).
- Line 420-540: This section seems a little redundant, please condense this section.
The text has been shortened to a somewhat extent.
Reviewer 3 Report
In this study, the authors provided a comprehensive characterization of two alpha-CA2 mutants for their photosynthetic parameters and their growth. The writing is clear, although there are some grammar issues that need careful revision to avoid confusion.
My main concerns are statistical analysis. Tables and supplemental documents also need statistical analysis to indicate the significance of the difference. In Figure 5, of which a "greater resistance to the prolonged action" is claimed (lines 327-328), a statistical analysis is also needed. In Figure 8, please indicate how many replicates were analyzed.
Some figures need to be revised to fit the style of the journal or of a formal publication, such as Figures 1-4, etc.
A major concern of mine is the last sentence in the Introduction section. "permitting to (make a) guess that α-CA2 is located in chloroplasts, most likely in thylakoid membranes". From the results of this study, there is no data to support this guess, although it looks like the knockout of alpha-CA2 affected the expression of beta-CA2 and 3. The related description should be removed from the manuscript.
Author Response
- My main concerns are statistical analysis. Tables and supplemental documents also need statistical analysis to indicate the significance of the difference. In Figure 5, of which a "greater resistance to the prolonged action" is claimed (lines 327-328), a statistical analysis is also needed. In Figure 8, please indicate how many replicates were analyzed.
For the data presented in Figure 5, in all tables and in supplemental documents, a statistical analysis was carried out. The results presented in Figure 8 were obtained from two independent experiments.
- Some figures need to be revised to fit the style of the journal or of a formal publication, such as Figures 1-4, etc.
Figures 1, 2 and 7 have been modified to match the style of the other figures.
- A major concern of mine is the last sentence in the Introduction section. "permitting to (make a) guess that α-CA2 is located in chloroplasts, most likely in thylakoid membranes". From the results of this study, there is no data to support this guess, although it looks like the knockout of alpha-CA2 affected the expression of beta-CA2 and 3. The related description should be removed from the manuscript.
The esteemed reviewer incorrectly indicated the effect of the knockout of alpha-CA2 on the expression levels of genes encoding beta-CA2 and beta-3. It is clearly seen in Fig. 8 that the expression levels of these genes did not change in α-CA2-KO as compared with their levels in WT plants.
We can’t agree with reviewer’s assertion that the results of our study do not support the guess about presence of alpha-CA2 in chloroplasts, and possibly in the thylakoid membrane. At first, the changes of the photosynthetic electron transport and the redox state of the plastoquinone pool in intact leaves (the latter one was estimated using two approaches with use of chlorophyll a fluorescence measurements) in α-CA2-KO as compared with WT plants already impelled us to suspect such location of alpha-CA2. The clear changes of the characteristics of electrochromic shift in intact leaves in the mutant as compared with WT plants demonstrates that this phenomenon, which occurs just in thylakoid membranes affected due to absence of α-CA2. The changes in electron transport rate measured in isolated thylakoids revealed the changes, which corresponded exactly to data, obtained measuring ECS in leaves. These data gave possibility to one of reviewers to note “The increase in the electron transport rate in isolated thylakoids and the ECS observed strongly indicate the association of the alpha-CA2 with the thylakoid membrane.” We may add that the observed influence of α-CA2 absence on state transitions, the processes taking place in thylakoid membrane and depending on the events occurring here can be also considered as indication of location of alpha-CA2 in thylakoid membrane, and, introducing the rather complicated propositions at least in chloroplasts. The changes in starch biosynthesis as well as the effect of alpha-CA2 absence on the expression levels of only chloroplast's CAs, of course very indirect evidences, although in combination with above ones also could be considered in favor of our guess.